# Training Normalizing Flows with the Information Bottleneck for Competitive Generative Classification

**Lynton Ardizzone\***
Visual Learning Lab, Heidelberg University
`lynton.ardizzone@iwr.uni-heidelberg.de`

**Radek Mackowiak\***
Bosch Center for Artificial Intelligence (CA)
`radek.mackowiak@gmail.com`

*\* Equal contribution*

**Carsten Rother**
Visual Learning Lab, Heidelberg University

**Ullrich Köthe**
Visual Learning Lab, Heidelberg University

## Abstract

The Information Bottleneck (IB) objective uses information theory to formulate a task-performance versus robustness trade-off. It has been successfully applied in the standard discriminative classification setting. We pose the question whether the IB can also be used to train generative likelihood models such as normalizing flows. Since normalizing flows use invertible network architectures (INNs), they are information-preserving by construction. This seems contradictory to the idea of a bottleneck. In this work, firstly, we develop the theory and methodology of IB-INNs, a class of conditional normalizing flows where INNs are trained using the IB objective: Introducing a small amount of *controlled* information loss allows for an asymptotically exact formulation of the IB, while keeping the INN's generative capabilities intact. Secondly, we investigate the properties of these models experimentally, specifically used as generative classifiers. This model class offers advantages such as improved uncertainty quantification and out-of-distribution detection, but traditional generative classifier solutions suffer considerably in classification accuracy. We find the trade-off parameter in the IB controls a mix of generative capabilities and accuracy close to standard classifiers. Empirically, our uncertainty estimates in this mixed regime compare favourably to conventional generative and discriminative classifiers. Code: `github.com/VLL-HD/IB-INN`

## 1   Introduction

The Information Bottleneck (IB) objective (Tishby et al., 2000) allows for an information-theoretic view of neural networks, for the setting where we have some observed input variable $X$, and want to predict some $Y$ from it. For simplicity, we limit the discussion to the common case of discrete $Y$ (i.e. class labels), but results readily generalize. The IB postulates existence of a latent space $Z$, where all information flow between $X$ and $Y$ is channeled through (hence the method's name). In order to optimize predictive performance, IB attempts to maximize the mutual information $I(Y, Z)$ between $Y$ and $Z$. Simultaneously, it strives to minimize the mutual information $I(X, Z)$ between $X$ and $Z$, forcing the model to ignore irrelevant aspects of $X$ which do not contribute to classification performance and only increase the potential for overfitting. The objective can thus be expressed as

$$\mathcal{L}_{\mathrm{IB}} = I(X, Z) - \beta \, I(Y, Z) \,. \tag{1}$$

The trade-off parameter $\beta$ is crucial to balance the two aspects. The IB was successfully applied in a variational form (Alemi et al., 2017; Kolchinsky et al., 2017) to train feed-forward classification models $p(Y|X)$ with higher robustness to overfitting and adversarial attacks than standard ones.

In this work, we consider the relationship between $X$ and $Y$ from the opposite perspective – using the IB, we train an invertible neural network (INN) as a conditional generative likelihood model

$p(X|Y)$, i.e. as a specific type of conditional normalizing flow. In this case, $X$ is the variable of which the likelihood is predicted, and $Y$ is the class condition. It is a generative model because one can sample from the learned $p(X|Y)$ at test time to generate new examples from any class, although we here focus on optimal likelihood estimation for existing inputs, not the generating aspect.

We find that the IB, when applied to such a likelihood model $p(X|Y)$, has special implications for the use as a so-called generative classifier (GC). GCs stand in contrast to standard *discriminative* classifers (DCs), which directly predict the class probabilities $p(Y|X)$. For a GC, the posterior class probabilities are indirectly inferred at test time by Bayes' rule, cf. Fig. 1: $p(Y|X) = p(X|Y)p(Y)/\mathbb{E}_{p(Y)}[p(X|Y)]$. Because DCs optimize prediction performance directly, they achieve better results in this respect. However, their models for $p(Y|X)$ tend to be most accurate near decision boundaries (where it matters), but deteriorate away from them (where deviations incur no noticeable loss). Consequently, they are poorly calibrated (Guo et al., 2017) and out-of-distribution data can not be easily recognized at test time (Ovadia et al., 2019). In contrast, GCs model full likelihoods $p(X|Y)$ and thus implicitly full posteriors $p(Y|X)$, which leads to the opposite behavior – better predictive uncertainty at the price of reduced accuracy. Fig. 2 illustrates the decision process in latent space $Z$.

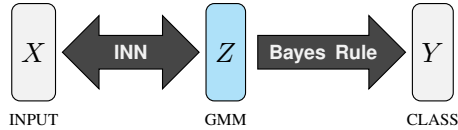

Figure 1: The Information Bottleneck Invertible Neural Network (IB-INN) as a generative classifier.

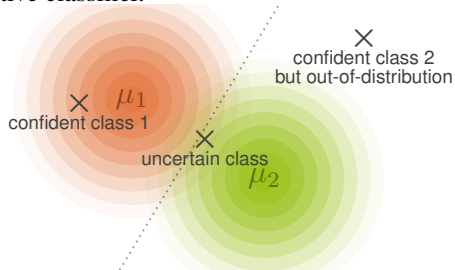

Figure 2: Illustration of the latent output space of a generative classifier. The two class likelihoods for $Y = \{1, 2\}$ are parameterized by their means $\mu_{\{1,2\}}$ in $Z$. The dotted line represents the decision boundary. A confident, an uncertain, and an out-of-distribution sample are illustrated.

In the past, deep learning models trained in a purely generative way, particularly flow-based models trained with maximum likelihood, achieved highly unsatisfactory accuracy, so that some recent work has called into question the overall effectiveness of GCs (Fetaya et al., 2019; Nalisnick et al., 2019b). In-depth studies of idealized settings (Bishop & Lasserre, 2007; Bishop, 2007) revealed the existence of a trade-off, controlling the balance between discriminative and generative performance. In this work, we find that the IB can represent this trade-off, when applied to generative likelihood models.

To summarize our contributions, we combine two concepts – the Information Bottleneck (IB) objective and Invertible Neural Networks (INNs). Firstly, we derive an asymptotically exact formulation of the IB for this setting, resulting in our IB-INN model, a special type of conditional normalizing flow. Secondly, we show that this model is especially suitable for the use as a GC: the trade-off parameter $\beta$ in the IB-INN's loss smoothly interpolates between the advantages of GCs (accurate posterior calibration and outlier detection), and those of DCs (superior task performance). Empirically, at the right setting for $\beta$, our model only suffers a minor degradation in classification accuracy compared to DCs while exhibiting more accurate uncertainty quantification than pure DCs or GCs.

## 2 Related Work

**Information Bottleneck:** The IB was introduced by Tishby et al. (2000) as a tool for information-theoretic optimization of compression methods. This idea was expanded on by Chechik et al. (2005); Gilad-Bachrach et al. (2003); Shamir et al. (2010) and Friedman et al. (2013). A relationship between IB and deep learning was first proposed by Tishby & Zaslavsky (2015), and later experimentally examined by Shwartz-Ziv & Tishby (2017), who use IB for the understanding of neural network behavior and training dynamics. A close relation of IB to dropout, disentanglement, and variational autoencoding was discovered by Achille & Soatto (2018), which led them to introduce Information Dropout as a way to take advantage of IB in discriminative models. The approximation of IB in a variational setting was proposed independently by Kolchinsky et al. (2017) and Alemi et al. (2017), who especially demonstrate improved robustness against overfitting and adversarial attacks.
**Generative Classification:** An in-depth analysis of the trade-offs between discriminative and generative models was first performed by Ng & Jordan (2001) and was later extended by Bouchard &

Triggs (2004); Bishop & Lasserre (2007); Xue & Titterington (2010), who investigated the possibility of balancing the strengths of both methods via a hyperparameter, albeit for very simple models. GCs have been used more rarely in the deep learning era, some exceptions being application to natural language processing (Yogatama et al., 2017), and adversarial attack robustness (Li et al., 2019; Schott et al., 2019). However, Fetaya et al. (2019) found that conditional normalizing flows have poor discriminative performance, making them unsuitable as GCs. GCs should be clearly distinguished from so-called hybrid models (Raina et al., 2004): these commonly only model the marginal $p(X)$ and jointly perform discriminate classification using shared features, with their main application being semi-supervised learning. Notable examples are Kingma et al. (2014); Chongxuan et al. (2017); Nalisnick et al. (2019c); Grathwohl et al. (2019).

## 3   Method

Below, upper case letters denote random variables (RVs) (e.g. $X$) and lower case letters their instances (e.g. $x$). The probability density function of a RV is written as $p(X)$, the evaluated density as $p(x)$ or $p(X{=}x)$, and all RVs are vector quantities. We distinguish true distributions from modeled ones by the letters $p$ and $q$, respectively. The distributions $q$ always depend on model parameters, but we do not make this explicit to avoid notation clutter. Assumption 1 in the appendix provides some weak assumptions about the domains of the RVs and their distributions. Full proofs for all results are also provided in the appendix.

Our models have two kinds of learnable parameters. Firstly, an invertible neural network (INN) with parameters $\theta$ maps inputs $X$ to latent variables $Z$ bijectively: $Z = g_\theta(X) \Leftrightarrow X = g_\theta^{-1}(Z)$. Assumption 2 in the Appendix provides some explicit assumptions about the network, its gradients, and the parameter space, which are largely fulfilled by standard invertible network architectures, including the affine coupling architecture we use in the experiments. Secondly, a Gaussian mixture model with class-dependent means $\mu_y$, where $y$ are the class labels, and unit covariance matrices is used as a reference distribution for the latent variables $Z$:

$$q(Z \,|\, Y) = \mathcal{N}(\mu_y, \mathbb{I}) \quad \text{and} \quad q(Z) = \sum_y p(y)\,\mathcal{N}(\mu_y, \mathbb{I}). \tag{2}$$

For simplicity, we assume that the label distribution is known, i.e. $q(Y) = p(Y)$. Our derivation rests on a quantity we call *mutual cross-information $CI$* (in analogy to the well-known cross-entropy):

$$CI(U, V) = \mathbb{E}_{u,v \sim p(U,V)} \left[ \log \frac{q(u,v)}{q(u)q(v)} \right]. \tag{3}$$

Note that the expectation is taken over the true distribution $p$, whereas the logarithm involves model distributions $q$. In contrast, plain mutual information uses the same distribution in both places. Our definition is equivalent to the recently proposed predictive $\mathcal{V}$-information (Xu et al., 2020), whose authors provide additional intuition and guarantees. The following proposition (proof in Appendix) clarifies the relationship between mutual information $I$ and $CI$:

**Proposition 1.** *Assume that $q(.)$ can be chosen from a sufficiently rich model family (e.g. a universal density estimator, see Assumption 2). Then for every $\eta > 0$ there is a model such that $\big| I(U,V) - CI(U,V) \big| < \eta$ and $I(U,V) = CI(U,V)$ if $p(u,v) = q(u,v)$.*

We replace both mutual information terms $I(X,Z)$ and $I(Y,Z)$ in Eq. 1 with the mutual cross-information $CI$, and derive optimization procedures for each term in the following subsections.

### 3.1   INN-Based Formulation of the I(X,Z)-Term in the IB Objective

Estimation of the mutual cross-information $CI(X,Z)$ between inputs and latents is problematic for deterministic mappings from $X$ to $Z$ (Amjad & Geiger, 2018), and specifically for INNs, which are bijective by construction. In this case, the joint distributions $q(X,Z)$ and $p(X,Z)$ are not valid Radon-Nikodym densities and both $CI$ and $I$ are undefined. Intuitively, $I$ and $CI$ become infinite, because $p$ and $q$ have an infinitely high delta-peak at $Z = g_\theta(X)$, and are otherwise 0. For the IB to be applicable, some information has to be discarded in the mapping to $Z$, making $p$ and $q$ valid Radon-Nikodym densities. In contrast, normalizing flows rely on all information to be retained for optimal generative capabilities and density estimation.

Our solution to this seeming contradiction comes from the practical use of normalizing flows. Here, a small amount of noise is commonly added to dequantize $X$ (i.e. to turn discrete pixel values

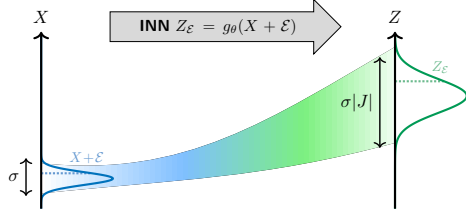

Figure 3: The more the noise is amplified in relation to the noise-free input, the lower the mutual cross-information between noisy latent vector $Z_\mathcal{E}$ and noise-free input $X$.

into real numbers), to avoid numerical issues during training. We adopt this approach to artificially introduce a minimal amount of information loss: Instead of feeding $X$ to the network, we input a noisy version $X' = X + \mathcal{E}$, where $\mathcal{E} \sim \mathcal{N}(0, \sigma^2\mathbb{I}) = p(\mathcal{E})$ is Gaussian with mean zero and covariance $\sigma^2\mathbb{I}$. For a quantization step size $\Delta X$, the additional error on the estimated densities caused by the augmentation has a known bound decaying with $\exp(-\Delta X^2/2\sigma^2)$ (see Appendix). We are interested in the limit $\sigma \to 0$, so in practice, we choose a very small fixed $\sigma$, that is smaller than $\Delta X$. This makes the error practically indistinguishable from zero. The INN then learns the bijective mapping $Z_\mathcal{E} = g_\theta(X + \mathcal{E})$, which guarantees $CI(X, Z_\mathcal{E})$ to be well defined. Minimizing this $CI$ according to the IB principle means that $g_\theta(X + \mathcal{E})$ is encouraged to amplify the noise $\mathcal{E}$, so that $X$ can be recovered less accurately, see Fig. 3 for illustration. If the global minimum of the loss is achieved w.r.t. $\theta$, $I$ and $CI$ coincide, as $CI(X, Z_\mathcal{E})$ is an upper bound (also cf. Prop. 1):

**Proposition 2.** *For the specific case that $Z_\mathcal{E} = g_\theta(X + \mathcal{E})$, it holds that $I(X, Z_\mathcal{E}) \leq CI(X, Z_\mathcal{E})$.*

Our approach should be clearly distinguished from applications of the IB to DCs, such as Alemi et al. (2017), which pursue a different goal. There, the model learns to ignore the vast majority of input information and keeps only enough to predict the class posterior $p(Y \mid X)$. In contrast, we induce only a small, explicitly adjustable loss of information to make the IB well-defined. As a result, the amount of retained information in our generative IB-INNs is orders of magnitude larger than in DC approaches, which is necessary to represent accurate class-conditional likelihoods $p(X \mid Y)$.

We now derive the loss function that allows optimizing $\theta$ and $\mu_y$ to minimize the noise-augmented $CI(X, Z_\mathcal{E})$ in the limit of small noise $\sigma \to 0$. Full details are found in appendix. We decompose the mutual cross-information into two terms

$$CI(X, Z_\mathcal{E}) = \mathbb{E}_{p(X),p(\mathcal{E})}\left[-\log q\big(Z_\mathcal{E} = g_\theta(x+\varepsilon)\big)\right] + \underbrace{\mathbb{E}_{p(X),p(\mathcal{E})}\left[\log q\big(Z_\mathcal{E} = g_\theta(x + \varepsilon) \,\big|\, x\big)\right]}_{:=A}.$$

The first expectation can be approximated by the empirical mean over a finite dataset, because the Gaussian mixture distribution $q(Z_\mathcal{E})$ is known analytically. To approximate the second term, we first note that the condition $X = x$ can be replaced with $Z = g_\theta(x)$, because $g_\theta$ is bijective and both conditions convey the same information

$$A = \mathbb{E}_{p(X),p(\mathcal{E})}\left[\log q\big(Z_\mathcal{E} = g_\theta(x + \varepsilon) \,\big|\, Z = g_\theta(x)\big)\right].$$

We now linearize $g_\theta$ by its first order Taylor expansion, $g_\theta(x + \varepsilon) = g_\theta(x) + J_x\varepsilon + O(\varepsilon^2)$, where $J_x = \frac{\partial g_\theta(X)}{\partial X}\big|_x$ denotes the Jacobian at $X = x$. Going forward, we write $O(\sigma^2)$ instead of $O(\varepsilon^2)$ for clarity, noting that both are equivalent because we can write $\varepsilon = \sigma n$ with $n \sim \mathcal{N}(0, \mathbb{I})$, and $\|\varepsilon\| = \sigma\|n\|$. Inserting the expansion into $A$, the $O(\sigma^2)$ can be moved outside of the expression: It can be moved outside the log, because that has a Lipschitz constant of $1/\inf q(g_\theta(X+\mathcal{E}))$, which we show is uniformly bounded in the full proof. The $O(\sigma^2)$ can then be exchanged with the expectation because the expectation's argument is also uniformly bounded, finally leading to

$$A = \mathbb{E}_{p(X),p(\mathcal{E})}\left[\log q\big(g_\theta(x) + J_x\varepsilon \,\big|\, g_\theta(x)\big)\right] + O(\sigma^2).$$

Since $\varepsilon$ is Gaussian with mean zero and covariance $\sigma^2\mathbb{I}$, the conditional distribution is Gaussian with mean $g_\theta(x)$ and covariance $\sigma^2 J_x J_x^T$. The expectation with respect to $p(\mathcal{E})$ is thus the negative entropy of a multivariate Gaussian and can be computed analytically as well

$$A = \mathbb{E}_{p(X)}\left[-\frac{1}{2}\log\big(\det(2\pi e\sigma^2 J_x J_x^T)\big)\right] + O(\sigma^2)$$

$$= \mathbb{E}_{p(X)}\left[-\log|\det(J_x)|\right] - d\log(\sigma) - \frac{d}{2}\log(2\pi e) + O(\sigma^2)$$

with $d$ the dimension of $X$. To avoid running the model twice (for $x$ and $x + \varepsilon$), we approximate the expectation of the Jacobian determinant by $0^{\text{th}}$-order Taylor expansion as

$$\mathbb{E}_{p(X)}\left[\log|\det(J_x)|\right] = \mathbb{E}_{p(X),p(\mathcal{E})}\left[\log|\det(J_\varepsilon)|\right] + O(\sigma),$$

where $J_\varepsilon$ is the Jacobian evaluated at $x + \varepsilon$ instead of $x$. The residual can be moved outside of the log and the expectation because $J_\varepsilon$ is uniformly bounded in our networks.

Putting everything together, we drop terms from $CI(X, Z_\mathcal{E})$ that are independent of the model or vanish with rate at least $O(\sigma)$ as $\sigma \to 0$. The resulting loss $\mathcal{L}_X$ becomes

$$\mathcal{L}_X = \mathbb{E}_{p(X),\, p(\mathcal{E})} \big[ -\log q\big(g_\theta(x+\varepsilon)\big) - \log \big| \det(J_\varepsilon) \big| \big]. \tag{4}$$

Since the change of variables formula defines the network's generative distribution as $q_X(x) = q\big(Z = g_\theta(x)\big) \big| \det(J_x) \big|$, $\mathcal{L}_X$ is the negative log-likelihood of the perturbed data under $q_X$,

$$\mathcal{L}_X = \mathbb{E}_{p(X),p(\mathcal{E})} \big[ -\log q_X(x + \varepsilon) \big]. \tag{5}$$

The crucial difference between $CI(X, Z_\mathcal{E})$ and $\mathcal{L}_X$ is the elimination of the term $-d \log(\sigma)$. It is huge for small $\sigma$ and would dominate the model-dependent terms, making minimization of $CI(X, Z_\mathcal{E})$ very hard. Intuitively, the fact that $CI(X, Z_\mathcal{E})$ diverges for $\sigma \to 0$ highlights why $CI(X, Z)$ is undefined for bijectively related $X$ and $Z$. In practice, we estimate $\mathcal{L}_X$ by its empirical mean on a training set $\{x_i, \varepsilon_i\}_{i=1}^N$ of size $N$, denoted as $\mathcal{L}_X^{(N)}$.

It remains to be shown that replacing $I(X, Z_\mathcal{E})$ with $\mathcal{L}_X^{(N)}$ in the IB loss Eq. 1 does not fundamentally change the solution of the learning problem in the limit of large $N$, small $\sigma$ and sufficient model power. Sufficient model power here means that the family of generative distributions realizable by $g_\theta$ should be a universal density estimator (see Appendix, Assumption 2). This is the case if $g_\theta$ can represent increasing triangular maps (Bogachev et al., 2005), which has been proven for certain network architectures explicitly (e.g. Jaini et al., 2019; Huang et al., 2018), including the affine coupling networks we use for the experiments (Teshima et al., 2020). Propositions 1 & 2 then tell us that we may optimize $CI(X, Z_\mathcal{E})$ as an estimator of $I(X, Z_\mathcal{E})$. The above derivation of the loss can be strengthened into

**Proposition 3.** *Under Assumptions 1 and 2, for any $\epsilon, \eta > 0$ and $0 < \delta < 1$ there are $\sigma_0 > 0$ and $N_0 \in \mathbb{N}$, such that $\forall N \geq N_0$ and $\forall 0 < \sigma < \sigma_0$, the following holds uniformly for all model parameters $\theta$:*

$$\Pr\left( \left| CI(X, Z_\mathcal{E}) + d \log \sqrt{2\pi e \sigma^2} - \mathcal{L}_X^{(N)} \right| > \epsilon \right) < \delta$$

$$and \quad \Pr\left( \left\| \frac{\partial}{\partial \theta} CI(X, Z_\mathcal{E}) - \frac{\partial}{\partial \theta} \mathcal{L}_X^{(N)} \right\| > \eta \right) < \delta$$

The first statement proves consistence of $\mathcal{L}_X^{(N)}$, and the second justifies gradient-descent optimization on the basis of $\mathcal{L}_X^{(N)}$. Proofs can be found in the appendix.

### 3.2  GMM-Based Formulation of the I(Z,Y)-Term in the IB Objective

Similarly to the first term in the IB-loss in Eq. 1, we also replace the mutual information $I(Y, Z)$ with $CI(Y, Z_\mathcal{E})$. Inserting the likelihood $q(z \,|\, y) = \mathcal{N}(z; \mu_y, \mathbb{I})$ of our latent Gaussian mixture model into the definition and recalling that $q(Y) = p(Y)$, this can be decomposed into

$$CI(Y, Z_\mathcal{E}) = \mathbb{E}_{p(Y)} \big[ -\log p(y) \big] + \mathbb{E}_{p(X,Y),p(\mathcal{E})} \left[ \log \frac{q\big(g_\theta(x+\varepsilon) \,|\, y\big)\, p(y)}{\sum_{y'} q\big(g_\theta(x+\varepsilon) \,|\, y'\big)\, p(y')} \right]. \tag{6}$$

In this case, $CI(Y, Z_\mathcal{E})$ is a lower bound on the true mutual information $I(Y, Z_\mathcal{E})$, allowing for its maximization in our objective. In fact, it corresponds to a bound originally proposed by Barber & Agakov (2003) (see their Eq. 3): The first term is simply the entropy $h(Y)$, because $p(Y)$ is known. The second term can be rewritten as the negative cross-entropy $-h_q(Y \mid Z_\mathcal{E})$. For $I(Y, Z_\mathcal{E})$, we would have the negative entropy $-h(Y \mid Z_\mathcal{E})$ in its place, then Gibbs' inequality leads directly to $CI(Y, Z_\mathcal{E}) \leq I(Y, Z_\mathcal{E})$.

The first expectation can be dropped during training, as it is model-independent. Note how the the second term can also be written as the expectation of the GMM's log-posterior $\log q(y \,|\, z)$. Since all mixture components have unit covariance, the elements of $Z$ are conditionally independent and the likelihood factorizes as $q(z \,|\, y) = \prod_j q(z_j \,|\, y)$. Thus, $q(y \,|\, z)$ can be interpreted as a naive Bayes classifier. In contrast to naive Bayes classifiers in data space, which typically perform badly because

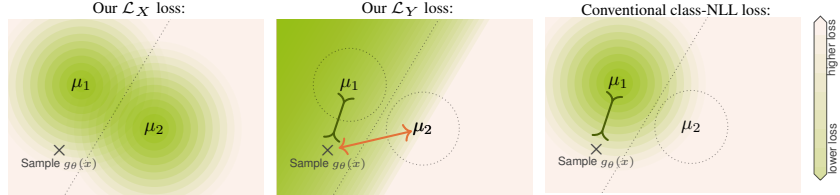

Figure 4: Illustration of the loss landscape for our IB formulation *(left, middle)* and standard class-conditional negative-log-likelihood *(right)*. The loss is shown for an input $x$ belonging to class $Y = 1$, green areas correspond to low loss. The orange arrows and black inverted arrows indicate repulsive and attractive interactions with the cluster centers. Crucially, standard NNL exerts no repulsive force.

raw features are not conditionally independent, our training enforces this property in latent space and ensures accurate classification. Defining the loss $\mathcal{L}_Y^{(N)}$ as the empirical mean of the log-posterior in a training set $\{x_i, y_i, \varepsilon_i\}_{i=1}^N$ of size N, we get

$$\mathcal{L}_Y^{(N)} = \frac{1}{N} \sum_{i=1}^{N} \log \frac{\mathcal{N}\big(g_\theta(x_i + \varepsilon_i); \mu_{y_i}, \mathbb{I}\big) \, p(y_i)}{\sum_{y'} \mathcal{N}\big(g_\theta(x_i + \varepsilon_i); \mu_{y'}, \mathbb{I}\big) \, p(y')}. \tag{7}$$

### 3.3 The IB-INN-Loss and its Advantages

Replacing the mutual information terms in Eq. 1 with their empirical estimates $\mathcal{L}_X^{(N)}$ and $\mathcal{L}_Y^{(N)}$, our model parameters $\theta$ and $\{\mu_1, ..., \mu_K\}$ are trained by gradient descent of the *IB-INN loss*

$$\mathcal{L}_{\text{IB-INN}}^{(N)} = \mathcal{L}_X^{(N)} - \beta \, \mathcal{L}_Y^{(N)} \tag{8}$$

In the following, we will interpret and discuss the nature of the loss function in Eq. 8 and form an intuitive understanding of why it is more suitable than the class-conditional negative-log-likelihood ('class-NLL') traditionally used for normalizing-flow type generative classifiers: $\mathcal{L}_{\text{class-NLL}} = -\mathbb{E} \log \big(q_\theta(x|y)\big)$. The findings are represented graphically in Fig. 4.

$\mathcal{L}_X$**-term:** As shown by Eq. 5, the term is the (unconditional) negative-log-likelihood loss used for normalizing flows, with the difference that $q(Z)$ is a GMM rather than a unimodal Gaussian. We conclude that this loss term encourages the INN to become an accurate likelihood model under the marginalized latent distribution and to ignore any class information.

$\mathcal{L}_Y$**-term:** Examining Eq. 7, we see that for any pair $(g(x + \varepsilon), y)$, the cluster centers $(\mu_{Y \neq y})$ of the other classes are repulsed (by minimizing the denominator), while $g_\theta(x + \varepsilon)$ and the correct cluster center $\mu_y$ are drawn together. Note that the class-NLL loss only captures the second aspect and lacks repulsion, resulting in a much weaker training signal. We can also view this in a different way: by substituting $q(x|y) \left| \det(J_x) \right|^{-1}$ for $q(z|y)$, the second summand of Eq. 6 simplifies to $\log q(y|x)$, since the Jacobian cancels out. This means that our $\mathcal{L}_Y$ loss directly maximizes the correct class probability, while ignoring the data likelihood. Again, this improves the training signal: as Fetaya et al. (2019) showed, the data likelihood will otherwise dominate the class-NLL loss, so that lack of classification accuracy is insufficiently penalized.

**Classical class-NLL loss:** The class-NLL loss or an approximation thereof is used to train standard GCs. The IB-INN loss reduces to this case for $\beta = 1$, because the first summand in $\mathcal{L}_X$ (cf. Eq. 4) cancels with the denominator in Eq. 7. Then, the INN no longer receives a penalty when latent mixture components overlap, and the GMM looses its class discriminatory power, as Fig. 4 illustrates: Points are only drawn towards the correct class, but there is no loss component repulsing them from the incorrect classes. As a result, all cluster centers tend to collapse together, leading the INN to effectively just model the marginal data likelihood (as found by Fetaya et al., 2019). Similarly, Wu et al. (2019) found that $\beta = 1$ is the minimum possible value to perform classification with discriminative IB methods.

## 4 Experiments

In the following, we examine the properties of the IB-INN used as a GC, especially the quality of uncertainty estimates and OoD detection. We construct our IB-INN by combining the design efforts of various works on INNs and normalizing flows. In brief, we use a Real-NVP architecture consisting of affine coupling blocks (Dinh et al., 2017), with added improvements from recent works (Kingma & Dhariwal, 2018; Jacobsen et al., 2019, 2018; Ardizzone et al., 2019). A detailed description of

the architecture is given in the appendix. We learn the set of means $\mu_Y$ as free parameters jointly with the remaining model parameters in an end-to-end fashion using the loss in Eq. 8. The practical implementation of the loss is explained in the appendix.

We apply two additional techniques while learning the model, label smoothing and loss rebalancing:

**Label smoothing** Hard labels force the Gaussian mixture components to be maximally separated, so they drift continually further apart during training, leading to instabilities. Label smoothing (Szegedy et al., 2016) with smoothing factor 0.05 prevents this, and we also apply it to all baseline models.

**Loss rebalancing** The following rebalancing scheme allows us to use the same hyperparameters when changing $\beta$ between 5 orders of magnitude. Firstly, we divide the loss $\mathcal{L}_X$ by the number of dimensions of $X$, which approximately matches its magnitude to the $\mathcal{L}_Y$ loss. We define a corresponding $\gamma := \beta/\dim(X)$ to stay consistent with the IB definition. Secondly, we scale the entire loss by a factor $2/(1+\gamma)$. This ensures that it keeps the same magnitude when changing $\gamma$.

$$\mathcal{L}_{\text{IB}}^{(N)} = \frac{2}{1+\gamma}\left(\frac{\mathcal{L}_X^{(N)}}{\dim(X)} - \gamma\,\mathcal{L}_Y^{(N)}\right) \tag{9}$$

Finally, the noise amplitude $\sigma$ should be chosen to satisfy two criteria: it should be small enough so that the Taylor expansions in the loss for $\sigma \to 0$ are sufficiently accurate, and it should also not hinder the model's performance. Our ablation provided in the Appendix indicates that both criteria are satisfied when $\sigma \lessapprox 0.25\Delta X$, with the quantization step size $\Delta X$, so we fix $\sigma = 10^{-3}$ for the remaining experiments.

## 4.1 Comparison of Methods

In addition to the IB-INN, we train several alternative methods. For each, we use exactly the same INN model, or an equivalent feed-forward ResNet model. Every method has the exact same hyperparameters and training procedure, the only difference being the loss function and invertibility.

**Class-NLL:** As a standard generative classifier, we firstly train an INN with a GMM in latent space naively as a conditional generative model, using the class-conditional maximum likelihood loss. Secondly, we also train a regularized version, to increase the classification accuracy. The regularization consists of leaving the class centroids $\mu_Y$ fixed on a hyper-sphere, forcing some degree of class-separation.

**Feed-forward** As a DC baseline, we train a standard ResNet (He et al., 2016) with softmax cross entropy loss. We replace each affine coupling block by a ResNet block, leaving all other hyperparameters the same.

**i-RevNet** (Jacobsen et al., 2018): To rule out any differences stemming from the constraint of invertibility, we additionally train the INN as a standard softmax classifier, by projecting the outputs to class logits. While the architecture is invertible, it is not a generative model and trained just like a standard feed-forward classifier.

**Variational Information Bottleneck (VIB):** To examine which observed behaviours are due to the IB in general, and what is specific to GCs, we also train the VIB (Alemi et al., 2017), a feed-forward DC, using a ResNet. We convert the authors definition of $\beta$ to our $\gamma$ for consistency.

## 4.2 Quantitative measurements

In the following, we describe the scores used in Table 1.

**Bits/dim:** The bits/dim metric is common for objectively comparing the performance of density estimation models such as normalizing flows, and is closely related to the KL divergence between real and estimated distributions. Details can be found e.g. in Theis et al. (2015).

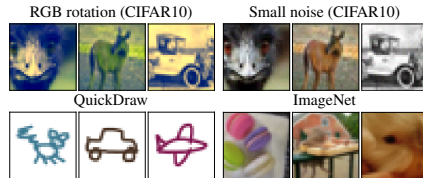

Figure 5: Examples from each OoD dataset used in the evaluation. The inlier data are original CIFAR10 images.

**Calibration error:** The calibration curve measures whether the confidence of a model agrees with its actual performance. All prediction outputs are binned according to their predicted probability $P$ ('*confidence*'), and it is recorded which fraction of predictions in each bin was correct, $Q$. For a perfectly calibrated model, we have $P = Q$, e.g. predictions with 70% confidence are correct 70% of the time. We use several metrics to measure deviations from this behaviour, largely in line with Guo et al. (2017). Specifically, we consider the expected calibration error (ECE, error weighted by bin count), the maximum calibration error (MCE, max error over all bins), and the integrated calibration error (ICE, summed error per bin), as well as the geometric

Table 1: Results on the CIFAR10 dataset. All models have the same number of parameters and were trained with the same hyperparameters. All values except entropy and overconfidence are given in percent. The arrows indicate whether a higher or lower value is better.

| Model | | Classif. err. (↓) | Bits/dim (↓) | Calibration error (↓) | | | | Incr. OoD prediction entropy (↑) | | | | | OoD detection score (↑) | | | | |
|---|---|---|---|---|---|---|---|---|---|---|---|---|---|---|---|---|---|
| | | | | Geo. mean | ECE | MCE | ICE | Average | RGB-rot | Draw | Noise | ImgNet | Average | RGB-rot | Draw | Noise | ImgNet |
| IB-INN (ours) | $\gamma = 1$ | 10.27 | 5.25 | **1.26** | **0.54** | **3.25** | **1.13** | **0.38** | **0.43** | 0.40 | **0.10** | **0.61** | 68.76 | **78.80** | 67.30 | 77.19 | 54.59 |
| | only $\mathcal{L}_X$ ($\gamma = 0$) | – | **4.80** | – | – | – | – | – | – | – | – | – | 74.51 | 70.68 | 85.74 | 91.14 | 55.82 |
| | only $\mathcal{L}_Y$ ($\gamma \to \infty$) | 8.72 | 17.27 | 3.98 | 0.81 | 13.94 | 5.57 | 0.28 | 0.23 | **0.40** | 0.00 | 0.49 | 61.25 | 57.04 | **90.29** | 50.24 | 54.40 |
| Stand. GC | Class-NLL | 61.75 | 4.81 | 12.61 | 4.17 | 30.58 | 15.70 | 0.03 | 0.02 | -0.06 | 0.02 | 0.12 | **73.92** | 70.65 | 83.31 | **90.97** | 55.76 |
| | Class-NLL + regul. | 40.04 | 4.83 | 24.75 | 7.13 | 70.63 | 30.11 | 0.01 | 0.00 | -0.01 | 0.01 | 0.02 | 74.02 | 69.33 | 85.13 | 91.04 | 55.88 |
| Pure DC | VIB ($\gamma = 1$) | 6.83 | – | 6.66 | 0.81 | 26.56 | 13.75 | 0.17 | 0.14 | 0.23 | 0.00 | 0.32 | – | – | – | – | – |
| | ResNet | **6.51** | – | 6.23 | 0.76 | 29.29 | 10.92 | 0.18 | 0.16 | 0.20 | 0.00 | 0.34 | – | – | – | – | – |
| | i-RevNet | 9.22 | – | 4.19 | 0.79 | 16.68 | 5.54 | 0.24 | 0.09 | 0.38 | 0.00 | 0.51 | – | – | – | – | – |

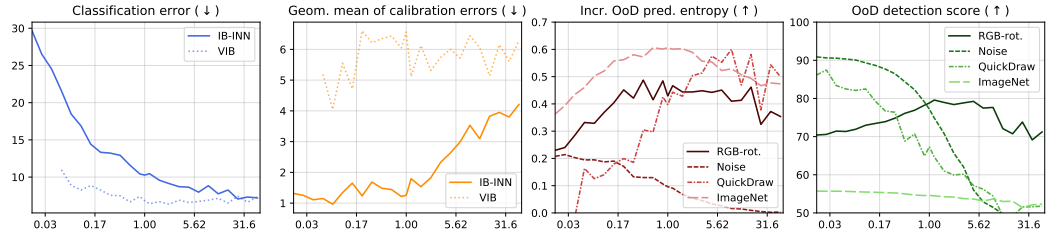

Figure 6: Effect of changing the parameter $\gamma$ between $0.02$ and $50$ (logarithmic $x$-axis) on the different performance measures ($y$-axis). The left two plots show the IB-INN and VIB, the right two plots only show the IB-INN. The VIB does not converge for $\gamma < 0.05$. The arrows indicate if a larger or smaller score is better. While classification accuracy improves with $\gamma$, the uncertainty measures generally grow worse. The trend of OoD detection and OoD entropy is less clear, and depends on the OoD dataset. The special case $\beta = 1$ (class-NLL) translates to $\gamma \approx 3 \cdot 10^{-4}$ (cf. Table 1).

mean of all three: $\sqrt[3]{\text{ECE} \cdot \text{MCE} \cdot \text{ICE}}$. The geometric mean is used because it properly accounts for the different magnitudes of the metrics. Exact definitions found in appendix.

**Increased out-of-distribution (OoD) prediction entropy:** For data that is OoD, we expect from a model that it returns uncertain class predictions, as it has not been trained on such data. In the ideal case, each class is assigned the same probability of $1/(\text{nr. classes})$. Ovadia et al. (2019) quantify this through the discrete entropy of the class prediction outputs $H(Y|X_{\text{Ood}})$. To counteract the effect of less accurate models having higher prediction entropy overall, we report the difference between OoD and in-distribution test set $H(Y|X_{\text{Ood}}) - H(Y|X_{\text{In distrib.}})$.

**OoD detection score:** We use OoD detection capabilities intrinsically built in to GCs. For this, we apply the recently proposed typicality test (Nalisnick et al., 2019a). This is a hypothesis test that sets an upper and lower threshold on the estimated likelihood, beyond which batches of inputs are classified as OoD. We apply the test to single input images (i.e. batch size 1). For quantification, we vary the detection threshold to produce a receiver operator characteristic (ROC), and compute the area under this curve (ROC-AUC) in percent. For short, we call this the *OoD detection score*. It will be 100 for perfectly separated in- and outliers, and 50 if each point is assigned a random likelihood.

**OoD datasets:** The inlier dataset consist of CIFAR10/100 images, i.e. $32 \times 32$ colour images showing 10/100 object classes. Additionally, we created four different OoD datasets, that cover different aspects, see Fig. 5. Firstly, we create a random 3D rotation matrix with a rotation angle of $\alpha = 0.3\pi$, and apply it to the RGB color vectors of each pixel of CIFAR10 images. Secondly, we add random uniform noise with a small amplitude to CIFAR10 images, as an alteration of the image statistics. Thirdly, we use the QuickDraw dataset of hand drawn objects (Ha & Eck, 2018), and filter only the categories corresponding to CIFAR10 classes and color each grayscale line drawing randomly. Therefore the semantic content is the same, but the image modality is different. Lastly, we downscale the ImageNet validation set to $32 \times 32$ pixels. In this case, the semantic content is different, but the image statistics are very similar to CIFAR10.

### 4.3 Results

**Quantitative Model Comparison** A comparison of all models is performed in Table 1 for CIFAR10, and in the appendix for CIFAR100. At the extreme $\gamma \to \infty$, the model behaves almost identically to a standard feed forward classifier using the same architecture (i-RevNet), and for $\gamma = 0$, it closely mirrors a conventionally trained GC, as the bits/dim are the same. We find the most favourable setting to be at $\gamma = 1$: Here, the classification error and the bits/dim each only suffer a 10% penalty compared to the extremes. The uncertainty quantification for IB-INN at this setting (calibration and

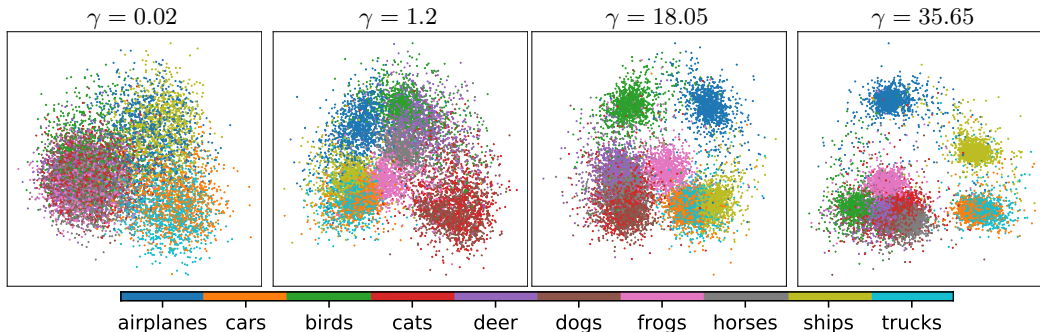

airplanes   cars   birds   cats   deer   dogs   frogs   horses   ships   trucks

Figure 7: GMM Latent space behaviour by increasing $\gamma$. The class separation increases with larger $\gamma$. Note that ambiguous classes (e.g. truck and car) remain connected to account for uncertainty.

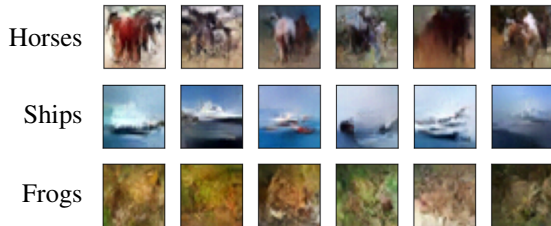

Figure 8: Images are generated for three different classes, by sampling from the respective mixture component in latent space, and inverting the network (more examples in Appendix). This gives insight what happens during classification, see text: only textures are generated for the frog class, indicating that this is the only aspect used for classification.

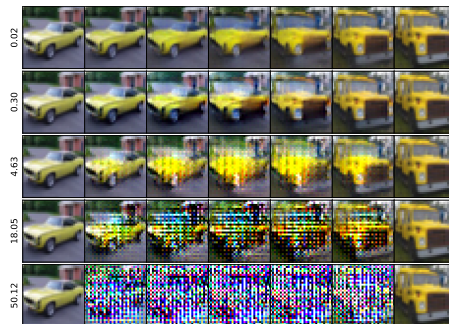

Figure 9: The columns show a latent space interpolation between two images (leftmost and rightmost). Each row shows a model with a different $\gamma$.

OoD prediction entropy) is far better than for pure DCs. Against expectations, standard GCs have worse calibration error. Our hypothesis is that their predictions are too noisy and inaccurate for a positive effect to be visible. For OoD detection, the IB-INN and standard GCs are all comparable, as we would expect from the similar bits/dim. Fig. 6 shows the trade-off between the two extremes in more detail: at low $\gamma$, the OoD detection and uncertainty quantification are improved, at the cost of classification accuracy. The VIB behaves in agreement with the other DCs: it has consistently lower classification error but higher calibration error than the IB-INN. This confirms that the IB-INN's behaviour is due to the application of IB to GCs exclusively. This does not mean that the IB-INN should be preferred over VIB, or vice versa. The main advantages of the VIB are the increased robustness to overfitting and adversarial attacks, aspects that we do not examine in this work.

**Latent Space Exploration** To better understand what the IB-INN learns, we analyze the latent space in different ways. Firstly, Fig. 7 shows the layout of the latent space GMM through a linear projection. We find that the clusters of ambiguous classes, e.g. truck and car, are connected in latent space, to account for uncertainty. Secondly, Fig. 9 shows interpolations in latent space between two test set images, using models trained with different values of $\gamma$. We observe that for low $\gamma$, the IB-INN has a well structured latent space, leading to good generative capabilities and plausible interpolations. For larger $\gamma$, class separation increases and interpolation quality continually degrades. Finally, generated images can give insight into the classification process, visualizing how the model understands each class. If a certain feature is not generated, this means it does not contribute positively to the likelihood, and in turn will be ignored for classification. Examples for this are shown in Fig. 8.

## 5 Conclusions

We addressed the application of the Information Bottleneck (IB) as a loss function to Invertible Neural Networks (INNs) trained as generative models. We find that we can formulate an asymptotically exact version of the IB, which results in an INN that is a generative classifier. From our experiments, we conclude that the IB-INN provides high quality uncertainties and out-of-distribution detection, while reaching almost the same classification accuracy as standard feed-forward methods on CIFAR10 and CIFAR100.

## Acknowledgements

LA received funding by the Federal Ministry of Education and Research of Germany project High Performance Deep Learning Framework (No 01IH17002). RM received funding from the Robert Bosch PhD scholarship. UK and CR received financial support from the European Research Council (ERC) under the European Unions Horizon 2020 research and innovation program (grant agreement No 647769). We thank the Center for Information Services and High Performance Computing (ZIH) at Dresden University of Technology for generous allocations of computation time. Furthermore we thank our colleagues (in alphabetical order) Tim Adler, Felix Draxler, Clemens Fruböse, Jakob Kruse, Titus Leistner, Jens Müller and Peter Sorrenson for their help and fruitful discussions.

## Broader Impact

As our IB-INN is not bound to any particular application, and applies to settings that can in principle already be solved with existing methods, we foresee no societal advantages or dangers in terms of direct application. More generally, we think accurate uncertainty quantification plays an important role in a safe and productive use of AI.

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
