[Supplementary Material]

# Training Normalizing Flows with the Information Bottleneck for Competitive Generative Classification – APPENDIX –

## Contents

## A Proofs and Derivations

### A.1 Assumptions

**Assumption 1.** *We assume that the the sample space $\mathcal{X}$ belonging to the input RV $X : \mathcal{X} \to \mathbb{R}$ is a compact domain in $\mathbb{R}^d$, and that $p(X \mid y)$ is absolutely continuous $\forall y \in \mathcal{Y}$, where $\mathcal{Y}$ is the set of available classes.*

The compactness of $\mathcal{X}$ is the major aspect here. However, this is always fulfilled for image data, as the pixels can only take certain range of values, and equally fulfilled for most other real-world datasets, as data representations, measurement devices, etc. only have a finite range.

**Assumption 2.** *We assume $g_\theta$ is from a family of universal density estimators, as defined by Definition 3 in Teshima et al. (2020). Moreover, we assume the network parameter space $\Theta$ is a compact subdomain of $\mathbb{R}^n$, $g_\theta$ and $J_\theta$ are uniformly bounded, and that the lower bound of $|\det J_\theta|$ is $> 0$. We also assume that $g_\theta$ and $J_\theta$ are continuous and differentiable in both $X$ and $\theta$.*

This is a fairly mild set of assumptions, as it is fulfilled by construction with most existing INN architectures using standard multi-layer subnetworks. See e.g. Behrmann et al. (2020); Virmaux & Scaman (2018) for details. Specifically, it holds for our tanh-clamped coupling block design (see

Appendix E). Note that some properties directly follow from Assumption 2: Firstly, as $J_\theta$ is uniformly bounded, this implies that $g_\theta$ is uniformly Lipschitz-continuous. Second, using Assumption 1, the domain of $Z = g_\theta(X)$ is compact, and $p(Z)$ is absolutely continuous.

## A.2 Mutual Cross-Information as Estimator for MI

In our case, we only require $CI(X, Z_\mathcal{E})$ and $CI(Y, Z_\mathcal{E})$, but we show the correspondence for two unspecified random variables $U, V$, as it may be of general interest. However, note that our estimator will likely not be particularly useful outside of our specific use-case, and other methods should be preferred (e.g. MINE, Belghazi et al., 2018). Our approach has the specific advantage, that we estimate the MI of the model using the model itself. For e.g. MINE, we would require three models, one generative model, and two models that only serve to estimate the MI. Secondly, it is not clear how the large constant $d \log(\sigma)$ can be cancelled out using other approaches.

For the joint input space $\Omega = \mathcal{U} \times \mathcal{V}$, we assume that $\mathcal{U}$ is a compact domain in $\mathbb{R}^d$, and $\mathcal{V}$ is either also a compact domain in $\mathbb{R}^l$ (Case 1), or discrete, i.e. a finite subset of $\mathbb{N}$ (Case 2). In Case 1, we assume that $p(U, V)$ is absolutely continuous with respect to the Lebesgue measure, and in Case 2, $p(U|v)$ is absolutely continuous for all values of $v \in \mathcal{V}$. This is in agreement with Assumption 1.

In Case 1, $q(U), q(V), q(U, V)$, the densities can all be modeled separately, by three flow networks $g_\theta^{(U)}(u), g_\theta^{(V)}(v), g_\theta^{(UV)}(u, v)$. Although in our formulation, we are later able to approximate the latter two through the first.

In Case 2, we only model $q(U|V)$, and assume that $q(V)$ is either known beforehand and set to $p(V)$ (e.g. label distribution), or the probabilities are parametrized directly. Either way, $q(U, V) = q(U|V)q(V)$ and $q(U) = \sum_{v \in \mathcal{V}} q(U, v)$.

**Proposition 1.** *Assume that the $q(.)$ densities can be chosen from a sufficiently rich model family (e.g. a universal density estimator). Then for every $\eta > 0$ there is a model such that*

$$\big|I(U, V) - CI(U, V)\big| < \eta \tag{1}$$

*and $I(U, V) = CI(U, V)$ if $p(U, V) = q(U, V)$.*

*Proof.* Writing out the definitions explicitly, and rearranging, we find

$$CI(U, V) = I(U, V) + D_{\mathrm{KL}}\big(p(U, V)\big\|q(U, V)\big)$$
$$- D_{\mathrm{KL}}\big(p(U)\big\|q(U)\big) - D_{\mathrm{KL}}\big(p(V)\big\|q(V)\big) \tag{2}$$

Shortening the KL terms to $D_1$, $D_2$ and $D_3$ for convenience:

$$|CI(U, V) - I(U, V)| = |D_1 - D_2 - D_3| \tag{3}$$
$$\leq D_1 + D_2 + D_3 \tag{4}$$
$$\leq 3 \max(D_1, D_2, D_3) \tag{5}$$

At this point, we can simply apply results from measure transport: if the $g_\theta$ are from a family of universal density estimators, we can choose $\theta^*$ to make $\max(D_1, D_2, D_3)$ arbitrarily small by matching $p$ and $q$. This was shown in general for increasing triangular maps, e.g. in Hyvärinen & Pajunen (1999), Theorem 1 for an accessible proof, or Bogachev et al. (2005) for a more in-depth approach (specifically Corollary 4.2). Generality was also proven for several concrete architectures, e.g. Teshima et al. (2020); Jaini et al. (2019); Huang et al. (2018).

For the second part of the Proposition, we note the following: if $p(U, V) = q(U, V)$, we have $D_1 = D_2 = D_3 = 0$, and therefore $CI(U, V) = I(U, V)$. $\qquad\square$

## A.3 Loss Function $\mathcal{L}_X$

In the following, we use the subscript-notation for the cross entropy:

$$h_q(U) = \mathbb{E}_{u \sim p(U)} \left[ -\log q(u) \right], \tag{6}$$

to avoid confusion with the joint entropy that arises with the usual notation ($h(p(U), q(U))$).

**Proposition 2.** *For the case given in the paper, that $Z_\mathcal{E} = g_\theta(X + \mathcal{E})$, it holds that $I(X, Z_\mathcal{E}) \leq CI(X, Z_\mathcal{E})$.*

*Proof.* In the following, we first use the invariance of the (cross-)information to homeomorphic transforms (see e.g. Cover & Thomas (2012) Sec. 8.6). Then, we use $p(X+\mathcal{E}|X) = q(X+\mathcal{E}|X) = p(\mathcal{E})$ (known exactly) and write out all the terms, most of which cancel. Finally, we use the inequality that the cross entropy is larger than the entropy, $h_q(U) \geq h(U)$ regardless of $q$. The equality holds iff the two distributions are the same.

$$CI(X, Z_\mathcal{E}) - I(X, Z_\mathcal{E}) = CI(X, X{+}\mathcal{E}) - I(X, X{+}\mathcal{E}) \tag{7}$$
$$= h_q(X) - h(X) + 0 \tag{8}$$
$$\geq 0 \tag{9}$$

With equality iff $p(X) = q(X)$. □

We now want to show that the network optimization procedure that arises from the empirical loss, in particular the gradients w.r.t. network parameters $\theta$, are consistent with those of $CI(X, Z_\mathcal{E})$:

**Proposition 3.** *The defined loss is a consistent estimator for $CI(X, Z_\mathcal{E})$ up to a known constant, and a consistent estimator for the gradients. Specifically, for any $\epsilon_1, \epsilon_2 > 0$ and $0 < \delta < 1$ there are $\sigma_0 > 0$ and $N_0 \in \mathbb{N}$, such that $\forall N \geq N_0$ and $\forall \sigma < \sigma_0$,*

$$\Pr\left( \left| CI(X, Z_\mathcal{E}) + d \log \sqrt{2\pi e \sigma^2} - \mathcal{L}_X^{(N)} \right| < \epsilon_1 \right) > 1 - \delta$$

*and*

$$\Pr\left( \left\| \frac{\partial}{\partial \theta} CI(X, Z_\mathcal{E}) - \frac{\partial}{\partial \theta} \mathcal{L}_X^{(N)} \right\| < \epsilon_2 \right) > 1 - \delta$$

*holds uniformly for all model parameters $\theta$.*

The loss function is as defined in the paper:

$$\mathcal{L}_X = h_q(Z_\mathcal{E}) - \mathbb{E}_{x \sim p(X+\mathcal{E})} \left[ \log \left| \det J_\theta(x) \right| \right] \tag{10}$$

as well as its empirical estimate using $N$ samples, $\mathcal{L}_X^{(N)}$.

We split the proof into two Lemmas, which we will later combine.

**Lemma 1.** *For any $\eta_1, \eta_2 > 0$ and $\delta > 0$ there is an $N_0 \in \mathbb{N}$ so that*

$$\Pr\left( \left| \mathcal{L}_X^{(N)} - \mathcal{L}_X \right| < \eta_1 \right) > 1 - \delta \tag{11}$$
$$\Pr\left( \left| \frac{\partial}{\partial \theta} \mathcal{L}_X^{(N)} - \frac{\partial}{\partial \theta} \mathcal{L}_X \right| < \eta_2 \right) > 1 - \delta \tag{12}$$
$$\forall N \geq N_0$$

*Proof.* For the first part (Eq. 11), we simply have to show that the uniform law of large numbers applies, specifically that all expressions in the expectations are bounded and change continuously with $\theta$. For the Jacobian term in the loss, this is the case by definition. For the $h_q(Z_\mathcal{E})$-term, we can show the boundedness of $\log q$ occurring in the expectation by inserting the GMM explicitly. We find

$$-\log(q(z)) \leq \max_y [(z - \mu_y)^2/2] + const. \tag{13}$$

while we know that $z = g_\theta(x)$ is bounded. Therefore, the uniform law of large numbers (Newey & McFadden, 1994, Lemma 2.4) guarantees existence of an $N_1$ to satisfy the condition for all $\theta \in \Theta$.

For the second part (Eq. 12), we will show that the gradient w.r.t. $\theta$ and the expectation can be exchanged, as the gradient is also bounded by the same arguments as before. We find that the conditions for exchanging expectation and gradient are trivially satisfied, again due to the bounded gradients (see L'Ecuyer (1995), assumption A1, with $\Gamma$ set to the upper bound). This results in an $N_2 \in \mathbb{N}$ for which Eq. 12 is satisfied. As a last step, we simply define $N_0 := \max(N_1, N_2)$. □

**Lemma 2.** *For any $\eta_1, \eta_2 > 0$ there is an $\sigma_0 > 0$, so that*

$$\left\| CI_\theta(X, Z_\mathcal{E}) + d\log\sqrt{2\pi e\sigma^2} - \mathcal{L}_X \right\| < \eta_2 \tag{14}$$

$$\left\| \frac{\partial}{\partial\theta}\Big(CI_\theta(X, Z_\mathcal{E}) - \mathcal{L}_X\Big) \right\| < \eta_2 \tag{15}$$

$$\forall \sigma < \sigma_0$$

*Proof.* In the following proof, we make use of the $O(\cdot)$ notation, see e.g. De Bruijn (1981):

We write $f(\sigma) = O(g(\sigma))$ $(\sigma \to 0)$ iff there exists a $\sigma_0$ and an $M \in \mathbb{R}$, $M > 0$ so that

$$\|f(\sigma)\| < M\,g(\sigma) \quad \forall \sigma \leq \sigma_0. \tag{16}$$

Furthermore, to discuss the limit case, it is necessary we reparametrize the noise variable $\mathcal{E}$ in terms of noise $S$ with a fixed standard normal distribution:

$$\mathcal{E} = \sigma S \quad \text{with} \quad p(S) = \mathcal{N}(0, 1) \tag{17}$$

To begin with, we use the invariance of $CI$ under the homeomorphic transform $g_\theta$. This can be easily verified by inserting the change-of-variables formula into the definition. See e.g. Cover & Thomas (2012) Sec. 8.6. This results in

$$CI(X, Z_\mathcal{E}) = CI(Z, Z_\mathcal{E}) = h_q(Z_\mathcal{E}) - h_q(Z_\mathcal{E}|Z) \tag{18}$$

Next, we series expand $Z_\mathcal{E}$ around $\sigma = 0$. We can use Taylor's theorem to write

$$Z_\mathcal{E} = Z + J_\theta(Z)\mathcal{E} + O(\sigma^2) \tag{19}$$

We have written the Jacobian dependent on $Z$, but note that it is still $\partial g_\theta/\partial X$, and we simply substituted the argument. We put this into the second entropy term $h_q(Z_\mathcal{E}|Z)$ in Eq. 18, and then perform a zero-order von Mises expansion of $h_q$. In general, the identity is

$$h_q(W + \xi) = h_q(W) + O(\|\xi\|) \quad (\|\xi\| \to 0), \tag{20}$$

and we simply put $\xi = O(\sigma^2)$ (the identity applies in the same way to the *conditional* cross-entropy). Intuitively, this is what we would expect: the entropy of an RV with a small perturbation should be approximately the same without the perturbation. See e.g. Serfling (2009), Sec. 6 for details. Effectively, this allows us to write the residual outside the entropy:

$$h_q(Z_\mathcal{E}|Z) = h_q\big(Z + J_\theta(Z)\mathcal{E} + O(\sigma^2)\big|Z\big) \tag{21}$$

$$= h_q\big(Z + J_\theta(Z)\mathcal{E}\big|Z\big) + O(\sigma^2) \tag{22}$$

$$= h_q\big(J_\theta(Z)\mathcal{E}\big|Z\big) + O(\sigma^2) \tag{23}$$

At this point, note that $q_\theta(J_\theta(Z)\mathcal{E}|Z)$ is simply a multivariate normal distribution, due to the conditioning on $Z$. In this case, we can use the entropy of a multivariate normal distribution, and simplify to obtain the following:

$$-h_q(J_\theta\mathcal{E}|Z) = \mathbb{E}\left[\frac{1}{2}\log\big(\det(2\pi\sigma^2 J_\theta J_\theta^T)\big)\right] \tag{24}$$

$$= \mathbb{E}\left[\frac{1}{2}\log\big((2\pi\sigma^2)^d \det(J_\theta)^2\big)\right] \tag{25}$$

$$= d\log\sqrt{2\pi e\sigma^2} + \mathbb{E}\left[\log|\det J_\theta|\right]. \tag{26}$$

Here, we exploited the fact that $J_\theta(Z)$ is an invertible matrix, and used $d = \dim(Z)$. Finally, as in practice we only want to evaluate the model once, we use the differentiability of $J_\theta$ to replace

$$\mathbb{E}\left[\log|\det J_\theta(Z)|\right] = \mathbb{E}\left[\log|\det J_\theta(Z_\mathcal{E})|\right] + O(\sigma). \tag{27}$$

The residual can be written outside of the expectation as we know it is bounded from our assumptions about $g_\theta$ and $J_\theta$ (Dominated Convergence theorem).

Putting the terms together, we obtain

$$CI(X, Z_{\mathcal{E}}) = h_q(Z_{\mathcal{E}}) - d \log \sqrt{2\pi e \sigma^2}$$
$$- \mathbb{E}\left[\log|\det J_\theta|\right] + O(\sigma) \tag{28}$$
$$= \mathcal{L}_X - d \log \sqrt{2\pi e \sigma^2} + O(\sigma) \tag{29}$$

Through the definition of $O(\cdot)$, Eq. 14 is satisfied. To show that the gradients also agree (Eq. 15), we must ensure that the $O(\sigma)$ term is uniformly convergent to 0 over $\theta$, i.e. there is a single constant $M$ in the definition of $O(\cdot)$ that applies for all $\theta \in \Theta$. This is directly the case, as $g_\theta$ is Lipschitz continuous and the outputs are bounded (Arzela - Ascoli theorem). $\qquad\square$

We can now combine the two Lemmas 1 and 2, to show Proposition 3.

**Proposition 3 - Proof.**

*Proof.* The Proposition follows directly from Lemmas 1 and 2: for a given $\epsilon_1$, $\epsilon_2$ and $\delta$, we choose each $\eta_i = \epsilon_i/2$, and apply the triangle inequality, meaning there exists an $N_0$ and $\sigma_0$ so that

$$\left| CI(X, Z_{\mathcal{E}}) + d \log \sqrt{2\pi e \sigma^2} - \mathcal{L}_X^{(N)} \right|$$
$$\leq \left| CI(X, Z_{\mathcal{E}}) + d \log \sqrt{2\pi e \sigma^2} - \mathcal{L}_X \right| + \left| \mathcal{L}_X - \mathcal{L}_X^{(N)} \right|$$
$$< \frac{\epsilon_1}{2} + \frac{\epsilon_1}{2}$$

And therefore $\Pr(\dots) > 1 - \delta$. Equivalently for the gradients.

$\qquad\square$

## A.4 Density Error through Noise Augmentation

For the derivation of the losses, we only assumed that $X$ and $X + \mathcal{E} =: X_{\mathcal{E}}$ are both RVs on a domain $\mathcal{X}$, and required no further assumptions about a possible quantization of $X$. However, *if* $X$ is quantized, which is mostly the case in practice, we can exploit this fact to derive a bound on the additional modeling error caused by the augmentation. To demonstrate this, we introduce the discrete, quantized data $W$. This is essentially the same as $X$, but is only defined on a finite, discrete set $\mathcal{W}$. With $F$ regular quantization steps in each of the $d$ dimensions, spaced by the quanitzation step size $\Delta X$, we write

$$\mathcal{W} = \left\{0, 1\Delta X, 2\Delta X \dots, (F-1)\Delta X\right\}^d \subset \mathcal{X}, \tag{30}$$

We denote probabilities of this discrete variable as upper case $P$ and $Q$ for true and modeled probabilities, respectively. We index the finite number of elements in $\mathcal{W}$ as $w_i$. For convenience, we also introduce the following notation:

$$P(w_i) =: P_i \qquad Q(w_i) =: Q_i. \tag{31}$$

Furthermore, we denote the noise distribution used for augmentation as $r(\mathcal{E})$ in the following, as this simplifies the notation and avoids ambiguities (it was denoted $p(\mathcal{E})$ instead for the loss derivation). From this, we can see how the distribution $p(X_{\mathcal{E}})$, which is used to train the network, can be expressed in terms of $P(W)$ and $r(\mathcal{E})$:

$$p(X_{\mathcal{E}}) = \sum_i P_i \, r(X_{\mathcal{E}} - w_i) \tag{32}$$

At test time, we want to recover an estimate $Q_i$. For standard normalizing flows, this is generally computed as

$$\tilde{Q}_i := \frac{q(X_{\mathcal{E}} = w_i)}{r(0)} \tag{33}$$

Among other things, this is used to measure the bits/dim. In the most general case, $\tilde{Q}$ will not sum to 1, so it is not guaranteed to be a valid probability, indicated by the tilde. Nevertheless, we can

see why this definition is sensible by considering the noise distribution $r$ used by most normalizing flows: hereby the support of $r$ in each dimension is smaller or equal to the quantization step size. Then, only one term in the sum in Eq. 32 is $\neq 0$ at any point. As a result, we obtain

$$q(X_\mathcal{E}) = p(X_\mathcal{E}) \implies \tilde{Q}(W) = P(W). \tag{34}$$

This means that in principle a standard normalizing flow can learn the true underlying discrete distribution from the noisy augmented distribution. In other words, the augmentation process does not introduce an additional error to the density estimation.

We now apply these definitions to our setting of a Gaussian noise distribution, $r(\mathcal{E}) = \mathcal{N}(0, \sigma^2 \mathbb{I})$. We consider the case where the model learns the training data distribution perfectly, i.e. $q(X_\mathcal{E}) = p(X_\mathcal{E})$. We find that Eq. 34 no longer holds for the Gaussian case, but that the error between $\tilde{Q}(W)$ and $P(W)$ has a known bound that decreases exponentially for small $\sigma$. For convenience, we write $A := \mathcal{N}(0; 0, \sigma^2 \mathbb{I}) = (2\pi\sigma^2)^{-d/2}$. From this, we get

$$\tilde{Q}_j = \frac{q(X_\mathcal{E} = w_j)}{A} = \frac{p(X_\mathcal{E} = w_j)}{A} \tag{35}$$

$$= \frac{1}{A} \sum_i P_i \mathcal{N}(w_j - w_i; 0, \sigma^2 \mathbb{I}) \tag{36}$$

$$= \frac{P_j \mathcal{N}(0; 0, \sigma^2 \mathbb{I})}{A} + \frac{1}{A} \sum_{i \neq j} P_i \mathcal{N}(w_j - w_i; 0, \sigma^2 \mathbb{I}) \tag{37}$$

$$= P_j + \underbrace{\frac{1}{A} \sum_{i \neq j} P_i \mathcal{N}(w_j - w_i; 0, \sigma^2 \mathbb{I})}_{:=\Delta P_j} \tag{38}$$

We are now interested in determining a bound for the error $\Delta P_j$. Because $\|w_i - w_j\| \geq \Delta X$ for $i \neq j$, we know

$$\mathcal{N}(w_i - w_j; 0, \sigma^2 \mathbb{I}) \leq A \exp\left(-\frac{\Delta X^2}{2\sigma^2}\right). \tag{39}$$

From that, we obtain the following bound:

$$\Delta P_j \leq \left(\sum_{i \neq j} P_i\right) \frac{1}{A} A \exp\left(-\frac{\Delta X^2}{2\sigma^2}\right) \tag{40}$$

$$\leq \exp\left(-\frac{\Delta X^2}{2\sigma^2}\right) \tag{41}$$

## B  Practical Loss Implementation

In the following, we provide the explicit loss implementations, as there are some considerations to make with regards to numerical tractability. Specifically, we make use of the operations `softmax`, `log_softmax`, `logsumexp` provided by major deep learning frameworks, as they avoid the most common pitfalls.

The class probabilities $q(Y)$ can be characterized through a vector $\Phi$, with

$$q(y) = \text{softmax}_y(\Phi), \tag{42}$$

where the subscript of the softmax operator denotes which index is selected for the enumerator. The use of the softmax ensures that $w_y$ stay positive and sum to one. For our work, $q(Y) = p(Y)$ is known beforehand, so we leave $\Phi$ fixed to 0 (equal probability for each class). However, we also find it is possible to learn $\Phi$ as a free parameter during training. In this case, only the gradients of the $\mathcal{L}_X$ loss w.r.t. $\Phi$ should be taken, as the $\mathcal{L}_Y$ loss is no longer a lower bound, and can be exploited by sending $\Phi_y \rightarrow \infty$ for some fixed $y$, and $\Phi_k \rightarrow -\infty$ for all $k \neq y$. If only $\mathcal{L}_X$ is backpropagated w.r.t. $\Phi$, this is avoided and $\Phi$ converges to the correct class weights. We use the shorthand $w_y := \log p(y)$ in the following.

With $z := g_\theta(x + \varepsilon)$, we also have

$$\bullet \log q(y) = w_y = \text{logsoftmax}_y(\Phi) \tag{43}$$

$$\bullet \log q(z|y) = -\frac{1}{2}\|z - \mu_y\|^2 + const. \tag{44}$$

$$\bullet \log q(z) = \text{logsumexp}_{y'}\left(-\frac{\|z - \mu_{y'}\|^2}{2} + w_{y'}\right) + const. \tag{45}$$

With this, the loss functions are evaluated as

$$\mathcal{L}_X(x) = \text{logsumexp}_{y'}\left(\frac{\|z - \mu_y'\|^2}{2} - w_{y'}\right) - \log J(x) \tag{46}$$

$$\mathcal{L}_Y(x, y) = \text{logsoftmax}_y\left(-\frac{\|z - \mu_{y'}\|^2}{2} + w_{y'}\right) - w_y. \tag{47}$$

The constants have been dropped for convenience. The use of the `logsumexp` and `logsoftmax` operations above is especially important. Otherwise when explicitly performing the exp and log operations with 32 bit floating point numbers, the values become too large, and the loss numerically ill-defined (`NaN`).

## C  Calibration Error Measures

In the following, we make use of the Iverson bracket:

$$[C] := \begin{cases} 1 & \text{if } C \text{ is true;} \\ 0 & \text{otherwise,} \end{cases} \tag{48}$$

Firstly, we define the bin edges $b_i$, with $i \in \{1, \ldots, K + 1\}$, so that $b_1 = 0$, $b_{K+1} = 1$, and $b_{i+1} > b_i$. In practice, we choose the $b_i$ be spaced more tightly near high and low confidences, as this is where the bulk of the predictions are made:

```
concatenate(range(0.00, 0.05, stepsize=0.01),
            range(0.05, 0.95, stepsize=0.1),
            range(0.95, 1.00, stepsize=0.01))
```

The bins themselves are then half-open intervals between the bin edges: $B_i = [b_i, b_{i+1})$ with $i \in \{1, \ldots, K\}$. We now define $n^{(i)}$, the count of predictions within a confidence bin; as well as $n_c^{(i)}$, the count of *correct* predictions in that bin:

$$n^{(i)} := \sum_{x_j} \sum_{y'} \left[p(y'|x_j) \in B_i\right] \tag{49}$$

$$n_c^{(i)} := \sum_{(x_j, y_j)} \sum_{y'} \left[p(y'|x_j) \in B_i\right] \cdot \left[\arg\max_{y'}(p(y'|x_j) = y_j\right] \tag{50}$$

where $x_j$ and the $(x_j, y_j)$-pairs are from the test set.

We define the confidence $P$ as the center of each bin, and the achieved accuracy in this bin as $Q$:

$$P_i = \frac{b_i + b_{i+1}}{2} \tag{51}$$

$$Q_i = \frac{n_c^{(i)}}{n^{(i)}} \tag{52}$$

Finally, using $Q$ and $P$, we define the calibration error measures, in agreement with Guo et al. (2017):

$$\text{ECE} = \sum_i \frac{n^{(i)}}{n_\text{tot}}|P_i - Q_i| \qquad \text{(Expected calib. err.)} \tag{53}$$

$$\text{MCE} = \max_i |P_i - Q_i| \qquad \text{(Maximum calib. err.)} \tag{54}$$

$$\text{ICE} = \sum_i (b_{i+1} - b_i)|P_i - Q_i| \qquad \text{(Integrated calib. err.)} \tag{55}$$

Figure 1: From left to right: Changes in test-loss, performance metrics, and a comparison between approximation and known slope of the true mutual information for varying values of $\sigma$ (x-axis)

Table 1: Results on the CIFAR100 dataset. All models have the same number of parameters and were trained with the same hyperparameters. All values except entropy and overconfidence are given in percent. The arrows indicate whether a higher or lower value is better.

| | Model | Classif. err. ($\downarrow$) | Bits/dim ($\downarrow$) | Calibration error ($\downarrow$) | | | | Incr. OoD prediction entropy ($\uparrow$) | | | | | OoD detection score ($\uparrow$) | | | | |
|---|---|---|---|---|---|---|---|---|---|---|---|---|---|---|---|---|---|
| | | | | Geo. mean | ECE | MCE | ICE | Average | RGB-rot | Draw | Noise | ImgNet | Average | RGB-rot | Draw | Noise | ImgNet |
| IB-INN (ours) | only $\mathcal{L}_X$ ($\gamma = 0$) | – | **4.82** | – | – | – | – | – | – | – | – | – | 70.03 | 63.35 | 87.45 | 85.12 | 50.99 |
| | $\gamma = 0.1$ | 42.57 | 4.94 | **2.60** | **0.58** | **7.04** | **4.28** | 0.50 | 0.66 | 0.28 | **0.35** | 0.69 | 68.31 | **66.53** | 78.91 | 81.70 | 50.75 |
| | only $\mathcal{L}_Y$ ($\gamma \to \infty$) | 33.78 | 18.44 | 4.49 | 0.62 | 16.76 | 8.72 | 0.58 | 0.52 | 1.04 | 0.00 | **0.77** | 58.29 | 47.95 | **99.37** | 49.23 | 49.23 |
| Stand. GC | Class-NLL | 97.92 | **4.82** | 16.20 | 1.02 | 95.63 | 43.53 | -0.04 | -0.14 | 0.55 | -0.53 | -0.03 | **70.26** | 64.68 | 86.54 | **85.19** | **51.09** |
| | Class-NLL + regul. | 69.28 | 5.07 | 13.94 | 0.75 | 89.74 | 40.15 | 0.00 | -0.00 | -0.01 | 0.01 | 0.01 | 68.83 | 64.96 | 82.19 | 83.32 | 50.44 |
| Stand. DC | ResNet | **29.27** | – | 5.13 | 0.65 | 20.57 | 10.16 | **0.60** | **0.68** | 0.97 | -0.00 | 0.74 | – | – | – | – | – |
| | i-RevNet | 37.54 | – | 5.18 | 0.63 | 19.85 | 11.09 | 0.51 | 0.32 | **1.00** | -0.00 | 0.75 | – | – | – | – | – |

using the shorthand $n_{\text{tot}} := \sum_i n^{(i)}$.

# D Additional Experiments

## D.1 Choice of $\sigma$

Fig. 1 shows the behaviour for 25 different models trained with $\sigma$ between $10^{-4}$ and $10^0$ (x-axis), and fixed $\gamma = 0.2$. We find that the loss values (left) and performance characteristics (middle) do not depend on $\sigma$ below a threshold that is about a factor 4 smaller than the qantization step size $\Delta X$. Contrary to expectations from existing work on normalizing flows, the models performance does not decrease even when $\sigma$ is 50 times smaller than $\Delta X$. Detrimental effects might occur more easily if the quantization steps are larger, e.g. $\Delta X = 1/32$ as used by Kingma & Dhariwal (2018), or if the model were more powerful or less regularized (e.g. from the tanh-clamping we employ). The rightmost plot compares our approximation of $CI(X, Z_\varepsilon)$ with the asymptotic $I(X, Z_\varepsilon) + const.$ for $\sigma \to 0$, where the constant is unknown. The slope of the approximation agrees well for small $\sigma$, but breaks down for larger values.

## D.2 CIFAR100

Table 1 reports the performance of the models on CIFAR100. The general behaviour observed for CIFAR10 is repeated here: The IB-INN model which balances both loss terms peforms significantly better in terms of uncertainty calibration than both standard GCs and DCs. It also performs OoD detection almost as well as pure GCs, with a much better classification error.

There are two differences compared to the CIFAR10 case: Firstly, in terms of increase in predictive entropy on OoD data, there are much smaller differences between models (excluding the standard GCs). The standard ResNet has the best overall performance by a small margin. Note that the increase in prediction entropy is also influenced by the calibration and overall classification error of the model to some degree, so we are careful in drawing any conclusions from minor differences. Secondly, we find that the most advantageous trade-off regime is now at a lower value of $\gamma$. The only values trained for CIFAR100 were $\gamma \in \{0.1, 1, 10\}$, and we find that the models with $\gamma$ set to 1 and 10 behave almost the same as the limit case $\gamma \to \infty$. The explanation for this is simple: due to the increased difficulty of the task, the $\mathcal{L}_Y$ loss is higher than for CIFAR10. Therefore, it has a larger influence at the same setting for $\gamma$ compared to the CIFAR10 models.

Figure 2: Effect of changing the parameter $\tilde{\beta}$ ($x$-axis) on the different performance measures ($y$-axis). The arrows indicate if a larger or smaller score is better. The black horizontal line in the last row indicates random performance. Details are explained in the paper. The VIB results are added as dotted lines. The VIB does not converge reliably for values of $\gamma < 0.2$, producing some otiliers e.g. for expected calibration error. This is not to claim that the IB-INN is better than the VIB or vice versa. The comparison serves to show how the IB affects GCs and DCs differently.

### D.3 Further experiments

Figure 2 provides all the performance metrics discussed in the paper over the range of $\gamma$.

In Figure 3 we show the trajectory of a sample in latent space, when gradually increasing the angle $\alpha$ of the RGB-rotation augmentation used in the paper as an OoD dataset. It travels from in-distribution to out-of-distribution. Such images were never seen during training.

Figure 5 shows samples generated by the model, using different values of $\gamma$. In general, we find the quality of generated images degrades faster with $\gamma$ than the interpolations between existing images. We see indications that the mass of points in latent space is offset from the learned $\mu_y$, meaning the regions that are sampled from have not seen much training data. In contrast to the IB-INN, the standard class-NLL trained model generates fairly generic looking images for all classes, due to the collapse of class-components in latent space.

## E   Network Architecture

As in previous works, our INN architecture consists of so-called *coupling blocks*. In our case, each block consists of one affine coupling (Dinh et al., 2017), illustrated in Fig. 4, followed by random and fixed soft permutation of channels (Ardizzone et al., 2019), and a fixed scaling by a constant, similar to ActNorm layers introduced by Kingma & Dhariwal (2018). For the coupling coefficients, each subnetwork predicts multiplicative and additive components jointly, as done by Kingma & Dhariwal (2018). Furthermore, we adopt the soft clamping of multiplication coefficients used by Dinh et al. (2017).

For downsampling blocks, we introduce a new scheme, whereby we apply the i-RevNet downsampling (Jacobsen et al., 2018) only to the inputs to the affine transformation ($u_2$ branch in Fig. 4), while the affine coefficients are predicted from a higher resolution $u_1$ by using a strided convolution in the corresponding subnetwork. After this, i-RevNet downsampling is applied to the other half of the channels $u_1$ to produce $v_1$, before concatenation and the soft permutation. We adopt this scheme

Figure 3: The scatter plot shows the location of test set data in latent space. A single sample is augmented by rotating the RGB color vector as described in the paper. The small images show the successive steps of augmentation, while the black arrow shows the position of each of these steps in latent space. We observe how the points in latent space travel further from the cluster center with increasing augmentation, causing them to be detected as OoD.

Forward computation (left to right):

$$v_1 = u_1, \ v_2 = T(u_2; nn(v_1))$$

Inverse computation (right to left):

$$u_1 = v_1, \ u_2 = T^{-1}(v_2; nn(u_1))$$

Figure 4: Illustration of a coupling block. $T$ represents some invertible transformation, in our case an affine transformation. The transformation coefficients are predicted by a subnetwork $(nn)$, which contains fully-connected or convolutional layers, nonlinear activations, batch normalization layers, etc., similar to the residual subnetwork in a ResNet (He et al., 2016). Note that how the subnetwork does not have to be inverted itself.

as it more closely resembles the standard ResNet downsampling blocks, and makes the downsampling operation at least partly learnable.

We then stack sets of these blocks, with downsampling blocks in between, in the manner of [8, down, 25, down, 25]. Note, we use fewer blocks for the first resolution level, as the data only has three channels, limiting the expressive power of the blocks at this level. Finally, we apply a discrete cosine transform to replace the global average pooling in ResNets, as introduced by Jacobsen et al. (2019), followed by two blocks with fully connected subnetworks.

We perform training with SGD, learning rate 0.07, momentum 0.9, and batch size 128, as in the original ResNet publication (He et al., 2016). We train for 450 epochs, decaying the learning rate by a factor of 10 after 150, 250, and 350 epochs.

Figure 5: Samples from four different models, each using the same architecture but a different loss. From top to bottom: class-NLL, $\gamma = 0.02$, $\gamma = 0.2$, $\gamma = 1$. In each subfigure, each row corresponds to one class. The classes from top to bottom are: plane, car, bird, cat, deer, dog, frog, horse, boat, truck.