[Reviews · NeurIPS 2020]

Review 1

Summary and Contributions: The paper introduces and formulizes an information bottleneck objective for use with invertible networks. Normally this would be impossible because invertible transformations are information preserving, so in order to get around this issue, they introduce an intermediate denoising procedure by adding small amounts of Gaussian noise. This fundamentally allows for an information bottleneck again. Technically, instead of targeting the IB objective directly, they target the V-information version of it, a sort of cross information where the expectations are taken with respect to real data while the models density is used to compute the log density ratios. For the particular instance presented here, this gives a particularly tractable form for the loss, involving a sort of jacobian smoothness constraint on the invertible mapping taking the place of the bottleneck. They show how their proposed objective / architecture behaves showing that they IB knob can smoothly explore a tradeoff between density modelling characteristics and discriminative classification performance with a seemingly sweet spot in between.

Strengths: I like the paper. I think its well written, the idea is novel, the work is honest. I feel as though I have learned something new in reading the paper and at its core is a new idea I wish I had thought of myself. To me that is precisely what a paper should provide. With particular interest of late in the field for invertible models and IB type methods, this paper does a good job of exploring their intersection, providing an objective that is arguably one of the only reasonable things one could define that is IB like for invertible models. I think the comparative experiments are well done and do illustrate not only the strengths but some of the weaknesses of the approach.

Weaknesses: So, while proposition 2 alleviates concerns about adopting the CI rather than I objective for X, Z_epsilon, the paper doesn't similarly address the replacement of I(z,y) with CI(z,y). I suspect you can not show that CI(Z, Y) <= I(Z,Y) (as you would want for the role it plays in the objective). If one cannot, I think this should be addressed and admitted in the paper, if one can than it certainly should be added. I would have liked to see more examples of generations from the model, because it relies crucially on the addition of small amounts of noise in pixel space, I suspect the model suffers quite a bit in its generative capabilities as it attempts to bottleneck tighter this seems true in Figure 8, where for the larger values of gamma the generations are all saturating, but I think this kind of discussion or admission or more demonstrations should be included in the text.

Correctness: I've spent some time checking the claims in the paper and haven't found any mistakes.

Clarity: Yes.

Relation to Prior Work: I believe so.

Reproducibility: Yes

Additional Feedback: Overall, the idea of using the data denoising procedure to give a consistent bottleneck is a clever one, and I do believe the claims and presentation in the paper is correct, I wonder if it isn't working against you a bit here. Zooming out for a minute, here we have the sort of Markov Chain Y -> Z -> X' -> (X, Epsilon), so that while we have an invertible mapping from Z to X', the X' itself is a gaussian perturbed X and that's where we get the bottlenecking. Now this ordinarily wouldn't be all that interesting since I(X; X + Epsilon) is just some data set dependent quantity that scales with sigma but isn't something model dependent (if the true distribution is used). The switch from MI to cross-MI here makes it so that we have model parameter control over CI(X; X+Epsilon), but here too, that control is somewhat limited. The only way the model can adjust the magnitude of the bottleneck is by making its X synthesizations more or less confusable by small amounts of Gaussian noise. That is, the only way it can adjust its bottleneck is to pay a large cost in its generative capability. In general small amounts of gaussian noise in pixel space are not particularly well aligned with the sorts of inductive biases we want on image datasets. I wonder if instead of adding the perturbation on the pixel side of the mapping, if it was added on the latent side things might not be a bit more interesting. The Z side of the flow is already gaussian like, and thus I suspect better aligned with small additive gaussian noise being a semantically interesting perturbation. I suspect in detail this might ruin the allaying of concerns that Proposition 2 provides, but might lead to better performing models in the end. Have the authors thought about putting the perturbation on the Z side of the map? Does it not work for some reason I'm failing to see? If they've thought about or tried it already I think adding this discussion to the paper would help future readers. ##### After Author Response Thank you for the comment about CI(Y,Z), you're right that if q(y) = p(y) which we assume is known then its just the variational lower bound again. I think that will help future readers to spell out in the paper. It's a nice paper and I'm happy it will have found a home.


Review 2

Summary and Contributions: This paper presents a technique for training normalizing flows. The paper looks really good but falls outside my field of expertise.

Strengths: Unfortunately, I can't judge the quality of this paper that falls outside the scope of my expertise. It looks technically sound, the equations make sense, and the quality of the figures is really appreciable.

Weaknesses: It could be nice to add a few sentences at the beginning of the paper explaining what the paper attempts to solve in layman's terms. I guess the final goal is to solve a classification problem, right?

Correctness: Probably.

Clarity: Yes, it is well written.

Relation to Prior Work: Literature review seems really exhaustive.

Reproducibility: Yes

Additional Feedback:


Review 3

Summary and Contributions: The paper combines the Information Bottleneck (IB) principle with Invertible Neural Nets (INN) in a completely new setting, competitive Generative Classification. While the standard motivation and application of the IB is supervised learning, generating efficient compressed representations, Z, of patterns (X) which maximize the information about their class (Y), given a sample of P(X,Y), this work originally propose and apply it to the opposite task - generative classification - find efficient generative representation Z that produce efficient class conditional generative models, p(X|Y), given the labeled sample. The paper is using a variational proxy to the Mutual Information, the cross information CI(X,Y), which the show to upper bound the mutual information under their GMM quantization of Z. They show that replacing the mutual Informations I(X;Z) and I(Z,Y) by CI(X;Z) and CI(Z;Y) in the IB tradeoff can form a useful practical variational proxy for the IB in high dimensions. On the other hand they use an invertible mapping from X to Z using an invertible NN (like i-RevNets) to generate a non-lossy representation Z, but turn it into a stochastic lossy map by adding Gaussian noise to each sample representation (the GMM’s). They then combine thieve two optimization problems, finding the best bijection from X to Z and then coarsen the representation by adding Gaussian noise using the IB proxy, to obtain Generative models that are in turned used for Generative Classification. The paper provides proofs for the validity of their bounds as proxy to the IB. In the experimental section, they compared the model to several natural alternatives, form simple Maximum Likelihood generative model (Class-NLL), feedforward Discriminative Classifier (DC), fully reversible network ((i-RevNet), and Variational Information Bottleneck (VIB), under carefully monitored similar conditions. The classification results are surprisingly in favor of the IB tradeoff coarsening where the patterns are blurred by a Gaussian noise following the trained non-linear bijection. The experimental results are diverse over several datasets and are quite compelling.

Strengths: This is an original (at least too me) application of the IB which is counterintuitive, as it works against the original motivation and most applications of the IB. It is combined nicely with the idea of invertible networks ((INN) which in turn enable the authors to prove rigorous Variational bounds when only Gaussian additive noise is used for the lossy compression Of the bottleneck. I found it rather elegant. The empirical evaluation is thorough and rather compelling, even that the actual classification test is weak and the detection score is marginal, the calibration errors are very convincing. Overall, this is an interesting paper that propose new method and application by combining first principles in a rather surprising way. The empirical tests are well executed and convincing.

Weaknesses: The papers is well written but the overall clarity can be improved (minor). I found the fact that the Cross Information is not always a bound on the mutual information - only in the additive Gaussian noise setting with differential entropies - somewhat disturbing. I would like to see a proxy to I(X;Y) that always bounds the mutual Information and obeys at least approximately the Data Processing Inequality, which is quite fundamental to the IB. I think these can be proved using the chain rules of the KL and actually obtain a stronger statement than prop 1, which as written is trivial. But these are minor and more theoretical critiques which don’t change much the quality of the paper.

Correctness: To the best of my knowledge the theoretical claims are correct, even if the proofs are somewhat too complicated (e.g. in prop 1) and some of the statements can actually be made stronger. The experiments seems well executed and the over presentation is nice - even if the graphics can be more esthetic.

Clarity: The paper is over all well written, but I think the mathematical propositions can be improved in clarity and simpler proofs. E.g, the statements about the proximity of the empirical derivatives rely on assumption on the bijective maps (such as smooth and bounded gradients) which are not mad explicitly enough in the proposition itself and may restrict the generality of the map. This is generally true when we combine Information measures with unrestricted 1-1 maps, but the maps be be restricted to be smooth enough. This fundamental technical difficulty is whipped under the rug and under some obscure mathematical terms both in the propositions and proofs. I’d like to see these made much more explicitly. I’m not convinced that the derivatives of the empirical cross information are good proxies to the derivatives of the expectation without these extra assumptions.

Relation to Prior Work: The paper is carefully referring to prior works both on the IB and INN. It’s different enough from everything I’ve seen in this domain.

Reproducibility: Yes

Additional Feedback:


Review 4

Summary and Contributions: [Update after author response: Thank you for your excellent response, especially the effect of hyperparameter sigma in the proposed framework. Though I still would like to know how can O(\sigma^2) be moved out of the log q(.) in Line 158-159, this is a small technical question and does not affect much the main conclusions of the paper. Thus, overall, I am happy to keep my score as it is.] The paper proposes a framework that connects Information Bottleneck principle into training of invertible neural networks (INN) by disturbing the input to the invertible map with a controlled noise. Various forms of approximation to the IB objective is derived for practical training objective in which an asymptotic bound for the approximations is also provided. The experimental evaluation illustrates the usefulness of the proposed framework to generative classifiers where the IB trade-off hyper-parameter controls the interpolation from being good uncertainty qualification and outlier detection to classification accuracy.

Strengths: There are several things to like about this paper: -A principled way of constructing an IB loss into INN with strong intuitions for each of the loss components. The derived loss generalises the standard INN loss. -An asymptotical analysis of the empirical loss in large sample and small noise magnitude limit. -Demonstrating an interesting applications for uncertainty estimates and OOD detection. -Extensive experiments with interesting results about interpolations between classification accuracy and generative capabilities

Weaknesses: However, I find the paper lack of a proper explanation/investigation for the role of \sigma in the generative-discriminative interpolation. \sigma is important in this problem setting because with \sigma = 0, the entire proposed framework is undefined. Thus, I think it is worthy to discuss/study the role of \sigma in controlling the generative-discriminative interpolation (beside the stability reason where \sigma makes the MI well-defined in this setting). Does such interpolation ability remains for a very small or very large value of \sigma?

Correctness: The technical content appears to be correct except some parts that might need better elaboration (see below) Line 158-159: O(\sigma^2) can be moved out the expectation, but before that, how O(\sigma^2) can be moved out of the log q(.) ? Line 164: The same question as above. In addition, should it be that J_\epsilon in the LHS and J_x in RHS? Line 184: should have been 0 < \sigma < \sigma_0?

Clarity: The paper is generally well-written except for some minor unclear parts (see below) Line 214-217: It is not immediately clear why such substitution results in log q(y|x). I think it would be better to elaborate on this in the paper. Line 157: O(\sigma^2) should have been O(\epsilon^2)

Relation to Prior Work: This work is well placed w.r.t prior work.

Reproducibility: Yes

Additional Feedback:

[Author Response · NeurIPS 2020]

We sincerely thank all reviewers for their time and constructive feedback. We will add all minor clarifications and
corrections to the final version (R2, R4, R5), as well as additional generated samples (R1). We are also thankful for the
idea of R1 on how to extend our method using perturbations on $Z$, and will investigate this in the future. We address the
main questions and criticisms in the following:

**$CI(Y, Z)$ as a lower bound (R1).** Thank you for this comment, we realized that we did not mention this in the
paper, even though the answer is straight forward and enlightening: $CI(Y, Z)$ is in fact a lower bound of $I(Y, Z)$.
This can be readily seen from Eq. 6: The first term is the entropy $h(Y)$, because the label distribution $p(Y)$ is known
exactly. The second term can be rewritten as the negative cross-entropy $-h_q(Y \mid Z)$. For $I(Y, Z)$, we have the
negative entropy $-h(Y \mid Z)$ as the second term instead. Because $h_q(Y \mid Z) \geq h(Y \mid Z)$ (Gibbs' inequality), we have
$CI(Y, Z) \leq I(Y, Z)$. This is essentially the the same as the variational bound originally proposed by Barber & Agakov
(2003): Their Eq. 3 corresponds to our Eq. 6, noting that their $x$ is our $Y$, and their $y$ is our $Z$. This is a bound that only
works in this specific case, as the label distribution $p(Y)$ must be known for it to apply. We will add this to the final
version.

**Mathematical assumptions about the network $g_{\theta}$ (R4).** We agree that the assumptions should be added to the text
and propositions more explicitly, and we will rectify this for the final version. However, we do not see the assumptions
as a 'fundamental technical difficulty': For all INN architectures used in practice, they are fulfilled by construction.
This includes GLOW, RealNVP, NICE, i-ResNet, and more. In none of these cases, there is any need for any additional
constraints, i.e. the assumptions are fulfilled per default. We refer to works such as Virmaux & Scaman (2018);
Behrmann et al. (2020) for further details.

**Strengthening Prop. 1 and properties of $CI$ (R4).** We also think that the $CI$ is of great interest in general and should
be further investigated in future. In our case, it is only used in a very specific way, so we did not consider strengthening
or extending Prop. 1. Instead, we would like to refer to Xu et al. (2020), who derive various further theoretical results
and insights concerning $CI$ in general.

**Effect of hyperparameter $\sigma$ (R5).** In line with this suggestion, we will add some more experiments to the appendix
concerning the effect of $\sigma$. As a first step, the following figure shows the behaviour for 25 different models trained with
$\sigma$ between $10^{-4}$ and $10^0$ (x-axis), and fixed $\gamma = 0.2$. We find that the loss values (left) and performance characteristics
(middle) do not depend on $\sigma$ below a threshold that is comparable to the qantization step size $\Delta X$. The models
performance does not decrease even when $\sigma$ is 50 times smaller than $\Delta X$. Detrimental effects might occur more
easily if the quantization steps are larger, e.g. $\Delta X = 1/32$ as used by Kingma & Dhariwal (2018). The rightmost plot
compares our approximation of $CI(X, Z_\varepsilon)$ with the asymptotic $I(X, Z_\varepsilon) + const.$ for $\sigma \to 0$, where the constant is
unknown. The slope of the approximation agrees well for small $\sigma$, but breaks down for larger values. This, and further
experiments concerning the role of $\sigma$ will be added to the final version.

34
**References.**

Barber, D. and Agakov, F. V. The im algorithm: a variational approach to information maximization. In *Advances in*
*neural information processing systems*, pp. None, 2003.
Behrmann, J., Vicol, P., Wang, K.-C., Grosse, R., and Jacobsen, J.-H. Understanding and mitigating exploding inverses
in invertible neural networks. *arXiv preprint arXiv:2006.09347*, 2020.
Kingma, D. P. and Dhariwal, P. Glow: Generative flow with invertible 1x1 convolutions. In *Advances in neural*
*information processing systems*, pp. 10215–10224, 2018.
Virmaux, A. and Scaman, K. Lipschitz regularity of deep neural networks: analysis and efficient estimation. In
*Advances in Neural Information Processing Systems*, pp. 3835–3844, 2018.
Xu, Y., Zhao, S., Song, J., Stewart, R., and Ermon, S. A theory of usable information under computational constraints.
*arXiv preprint arXiv:2002.10689*, 2020.


[Meta-Review · NeurIPS 2020]

High-quality paper, that demonstrates how the information-bottleneck principle can be adapted to train invertible neural networks, explores its theoretical and practical implications, and connects it to generative classification. All reviewers agree that the theory is interesting and novel. Well done. The reviews raised some interesting points (e.g. CI(X, Z) as a lower bound, the effect of sigma), which the rebuttal went on to address. I would encourage the authors to use the extra 9th page to add these details to the camera-ready too.